# Brain circuits activated by female sexual behavior evaluated by manganese enhanced magnetic resonance imaging

**Alejandro Aguilar-Moreno[1], Juan Ortiz[1], Luis Concha[1], Sarael Alcauter[1], Raúl G. Paredes[1,2]***

**1** Instituto de Neurobiología, UNAM, Campus Juriquilla, Querétaro, México, **2** Escuela Nacional de Estudios Superiores, Unidad Juriquilla, UNAM, Querétaro, México

* rparedes@unam.mx

**Data Availability Statement:** All relevant data are within the paper and its Supporting Information files.

## Abstract

Magnetic resonance imaging (MRI) allows obtaining anatomical and functional information of the brain in the same subject at different times. Manganese-enhanced MRI (MEMRI) uses manganese ions to identify brain activity, although in high doses it might produce neurotoxic effects. Our aims were to identify a manganese dose that does not affect motivated behaviors such as sexual behavior, running wheel and the rotarod test. The second goal was to determine the optimal dose of chloride manganese ($MnCl_2$) that will allow us to evaluate activation of brain regions after females mated controlling (pacing) the sexual interaction. To achieve that, two experiments were performed. In experiment 1 we evaluated the effects of two doses of $MnCl_2$, 8 and 16 mg/kg. Subjects were injected with one of the doses of $MnCl_2$ 24 hours before the test on sessions 1, 5 and 10 and immediately thereafter scanned. Female sexual behavior, running wheel and the rotarod were evaluated once a week for 10 weeks. In experiment 2 we followed a similar procedure, but females paced the sexual interaction once a week for 10 weeks and were injected with one of the doses of $MnCl_2$ 24 hours before the test and immediately thereafter scanned on sessions 1, 5 and 10. The results of experiment 1 show that neither dose of $MnCl_2$ induces alterations on sexual behavior, running wheel and rotarod. Experiment 2 demonstrated that MEMRI allow us to detect activation of different brain regions after sexual behavior, including the olfactory bulb (OB), the bed nucleus of the stria terminalis (BNST), the amygdala (AMG), the medial preoptic area (MPOA), the ventromedial hypothalamus (VMH), the nucleus accumbens (NAcc), the striatum (STR) and the hippocampus (Hipp) allowing the identification of changes in brain circuits activated by sexual behavior. The socio sexual circuit showed a higher signal intensity on session 5 than the reward circuit and the control groups indicating that even with sexual experience the activation of the reward circuit requires the activation of the socio sexual circuit. Our study demonstrates that MEMRI can be used repeatedly in the same subject to evaluate the activation of brain circuits after motivated behaviors and how can this activation change with experience.

**Funding:** This research was supported by Dirección General de Asuntos del Personal Académico (DGAPA, PAPIIT), Universidad Nacional Autónoma de México (Grant: DGAPA, PAPIIT, UNAM, IN206521). The funders had no role in study design, data collection and analysis, decision to publish, or preparation of the manuscript.

**Competing interests:** The authors have declared that no competing interests exist.

## Introduction

Sexual behavior (SB) is a motivated behavior that is fundamental for the survival of species. It is an innate behavior that can be triggered by hormones and chemo sexually relevant olfactory cues (pheromones) from conspecifics of the opposite sex. When a sexually receptive female rat is exposed to a male rat, the sexually relevant olfactory cues of the male are detected trough the olfactory system of the female. Under appropriate conditions, the female will display proceptive behaviors including hopping, darting and ear wiggling to attract the attention of the male. In response to a male mount the female will display a lordosis response to facilitate copulation [1].

Classic studies have shown that when females control or pace the sexual interaction it has positive effects in reproduction, increasing the probability of pregnancy and having a higher number of pups in the offspring compared to females that do not pace the sexual interaction [1, 2]. Paced mating also induces a reward state assuring that the behavior will be repeated in the future [3, 4]. It is also well established that paced mating induces neurogenesis in the subventricular zone (SVZ), rostral migratory stream (RMS), olfactory bulb system (OB) and the hippocampus indicating that this motivated behavior induces permanent brain plastic changes [5, 6].

The olfactory bulbs are part of a well identify circuit involved in the control of sexual behavior in rodent species. Pheromones are preferentially detected by the vomeronasal organ, which send its axons to the accessory olfactory bulb (AOB). From here, mitral cells project to the medial amygdala (AMG) which in turn projects to the bed nucleus of the stria terminalis (BNST) and the medial preoptic area (MPOA). The importance of this circuit in the control of reproduction including sexual behavior is well established [7, 8]. Other techniques such as hormone implants, lesions or immediate early gene expression have also demonstrated the importance of these brain regions in the control of sexual behavior. For example, the FOS protein was increased in the BNST, VMH and medial amygdala after sexually receptive females received vaginocervical stimulation [9]. A similar increase in FOS was observed in the BNST, the MPOA and the ventromedial hypothalamus (VMH) in females after receiving intromissions, compared to home-cage females [10]. Moreover, paced mating induced a higher FOS expression in the brain regions described above in comparison to females that do not paced the sexual interaction [11]. One of the limitations associated with some of the techniques used to identify the role of brain regions in the control of sexual behavior is that the animals need to be sacrificed after the behavior, for example to evaluate FOS expression [10]. The use of neuroimaging techniques allows us to test the same animal repeatedly and evaluate possible brain changes associated with experience in the same subject.

Magnetic resonance imaging (MRI) is a powerful technique that provides anatomical and functional brain information [12–14], being an excellent tool to perform longitudinal studies. MRI can measure anatomical and functional changes in the brain, including cortical thickness and brain neural activity [15–17]. Manganese-enhanced MRI (MEMRI) is a technique used as tracer of neural activity [16, 18–20]. MEMRI uses two characteristics of manganese: it is an analog of calcium, facilitating the entrance to depolarized cells through calcium voltage dependent channels [21]; and is a paramagnetic element, increasing MR contrast in T1 weighted images [13, 14, 22–24]. Hence, when the neurons are depolarized, the active cells increase T1 signal intensity by the accumulation of manganese. Therefore, the increase of signal intensity is related to the activation of neurons.

The first study to evaluate the functional properties of manganese was done by Lin & Koretsky [22]. Since then, MEMRI has been used in different animal models [17, 25–27] to evaluate brain circuits controlling different behaviors, using a wide range of doses, MR sequences

and image processing [18, 28–31]. It is well known that chloride manganese in high doses induce toxic effects, mainly in the motor skills of the subjects [25, 28–30, 32].

The aims of the present study were: 1) Find a dose of $MnCl_2$ that do not affect female sexual behavior, running wheel, and fine motor execution evaluated in the rotarod; 2) Identify the brain areas and/or circuits activated by female paced sexual behavior and determined if the activation is modified by sexual experience.

## Methods

### Subjects

Wistar female rats without sexual experience were used for this study. They were bilaterally ovariectomized (OVX) under anesthesia with a mixture of ketamine (95 mg/kg) and xylazine (12 mg/kg). Two weeks later and to induce sexual receptivity, OVX females received a subcutaneous (sc) injection of E2 benzoate (25 µg/rat; Sigma St. Louis, Missouri, USA) and progesterone (1 mg/rat; Sigma), 48 and 4 hours respectively, before the behavioral test. These doses were used to induced high levels of sexual receptivity and we have used them repeatedly before [33, 34]. For Sexual behavior tests, sexually experienced males were used as stimulus. Experiments were approved by the Instituto de Neurobiología Animal Care Committee which follows NIH guidelines for the use and care of animals.

**Experiment 1.** The aim of experiment 1 was to assess possible behavioral alterations induced by repeated manganese chloride ($MnCl_2$) sc administration in female rats. Aside from testing the effects on sexual behavior we also evaluated the effects of $MnCl_2$ in another motivated behavior, running wheel, and a forced motor task, the rotarod test. We chose the lowest doses reported in the literature that enhance the MR signal [35]. In a pilot study we tried 3 doses: 8, 16 and 32 mg/kg of chloride manganese. However, the subjects injected with the 32 mg/kg dose showed severe skin lesions in the site of the injection and reduced activity in the behavioral tests. Therefore, this dose was suspended. The animals were divided in the following groups: Control, injected with saline 24 hours before the behavioral test (saline, n = 9). The 8 mg/kg (n = 8) and 16 mg/kg (n = 9) groups were injected with their corresponding dose of $MnCl_2$ 24 hours before the behavioral tests. We evaluated the subjects in the following tests:

*Sexual Behavior Tests (SBT).* Each female was placed in a mating cage (40x60x40 cm) of polycarbonate for 30 minutes. To allow the females to pace the sexual interaction with the male, a division with a whole (7x5 cm) in the center was placed in the middle of the cage, making possible only for the females to move between the two sides of the cage. The size of the hole is sufficient to allow the female, but not the male, to move freely from one compartment to the other pacing the sexual interactions. The subjects were tested once a week for 10 weeks with sexually experienced males, registering the following parameters: number of mounts (M), intromissions (I), and ejaculations (E); latency to the first mount (ML), intromission (IL), and ejaculation (EL); inter-intromission interval (III), percent of exits and the return latencies after mounts, intromissions, and ejaculations. The percentage of exits is obtained by determined the number of times the female exits the male compartment after a mount, intromission, or ejaculation in comparison to the total number of each stimulus received. While the return latencies are calculated by obtaining the mean time for the female to return to the male's chamber after an exit following a mount, an intromission, or an ejaculation. The mean lordosis intensity (MLI: 0, no lordosis; 1, medium lordosis; 2, full lordosis), and the lordosis quotient (LQ; number of lordosis display in response to a mount by the male, divided by the number of mounts and multiplied by 100) were calculated. The sexual behavior tests lasted 30 minutes. If the stimulus male rat did no display sexual behavior in the first 10 minutes, it was replaced by another male.

*Running Wheel Test (RWT)*. Before the experiment, the females were trained in the running wheel in three sessions. The RW arena (LE904 76–0412, Panlab-Harvard Aparatus®) consist of a polycarbonate cage (42x26x19 cm, Allentown Caging Equipment) attached to a stainless-steel wheel activity (36x10 cm). A Multicounter (LE3806 Model) was used to record the number of turns, and the SEDACOM 2.0 (Panlab-Harvard Aparatus®) was used to export the data to a computer. The distance traveled was calculated using the circumference of the wheel (diameter $^*$ π, 0.36 m $^*$ 3.14 = 1.13 m) multiplied by the number of wheel turns (1.13 m $^*$ # of laps) and presented in meters (m). The duration of the RW training and test was 30 minutes for each session and subjects were tested once a week for 10 weeks immediately after the SBT.

*Rotarod Test (RT)*. The Rotarod Test (IITC, Inc. Life Science, Series 8) was used to assess motor coordination. Females were habituated and trained in five sessions before testing due the complexity of the task. Females were placed on top of the apparatus (cylinders with 9.5 cm of diameter, 30 cm above the surface of the apparatus) and the rotarod was programmed in two modalities, Regular Speed (RS) and Increase Speed (IS). In the regular speed modality, the rotarod rotated at 10 rpm for 180 seconds. During the IS modality the rotarod started at 10 rpm, increasing the speed until 15 rpm for a total duration of 120 s, followed by 60 s in that speed until the end of the test. In both tests the latency to the first fall was registered and the test ended.

*Chloride manganese (MnCl2) administration*. Chloride manganese ($MnCl_2$) (Manganese (II) chloride SIGMA-ALDRICH, Product number 244589, St. Louis) was dissolved in saline ($MnCl_2$mg/kg/10ml) and injected in doses of 8 or 16 mg/kg. In experiment 1 and 2, $MnCl_2$ was injected subcutaneously (sc) 24 hours before the behavioral evaluation [19] in weeks 1, 5 and 10. The doses were selected from previous studies [35].

*Behavioral testing procedure0*. Subjects were tested once a week for 10 weeks. They were tested for sexual behavior in 30 minutes test, immediately followed by RW for 30 minutes and then the rotarod test (Fig 1). Twenty-four hours before sessions 1, 5 and 10 subjects were injected with saline, or 8 or 16 mg/kg of $MnCl_2$.

**Experiment 2.** Once we determined that the doses of $MnCl_2$ tested did not produce non-specific behavioral effects we evaluated the contrast in MR images in females that mated pacing the sexual interaction compared to females that did not mate. Another set of OVX females was divided into the following groups. Females that did not mate were injected with 8 (n = 13) or 16 (n = 13) mg/kg of $MnCl_2$; Females that mated pacing the sexual interaction injected with 8 (n = 13) or 16 (n = 13) mg/kg of $MnCl_2$. We followed a procedure similar to experiment 1. Subjects in the paced mating groups were tested once a week for 10 weeks for sexual behavior (Fig 2). Females from the control groups were placed alone in the mating cage. Twenty-four hours before sessions 1, 5 and 10 subjects were injected with 8 or 16 mg/kg of $MnCl_2$. The sexual behavior tests were done as described in experiment 1.

*MEMRI data collection*. After the behavioral tests, the females were taken to the Laboratorio Nacional de Imagenología por Resonancia Magnética (LANIREM) for small species. The animals were anesthetized with isoflurane (Sofloran® Vet) at 4% for induction and 1.5–2% during the scan time. To maintain body temperature, circulating hot water was distributed under the base where the animal was placed inside the scanner. Body temperature and breathing frequency was monitored throughout the scan. To obtain the images a 7 Tesla scan (BioScan Bruker® PharmaScan) was used, with a transmit-receive volume coil and a head holder with adjustable bite to position the animal in the center of the magnet. The sequence used was a T1 weighted fast low angle shot (T1FLASH) with the following parameters: echo time, 8 ms; repetition time, 50 ms; number of acquisitions, 2; slice width, 16 mm; image size, 150x160x80; field of view, $30x32x16 mm^3$; flip angle, 20˚; total scan time of 23 min, 28 s. After the scanning

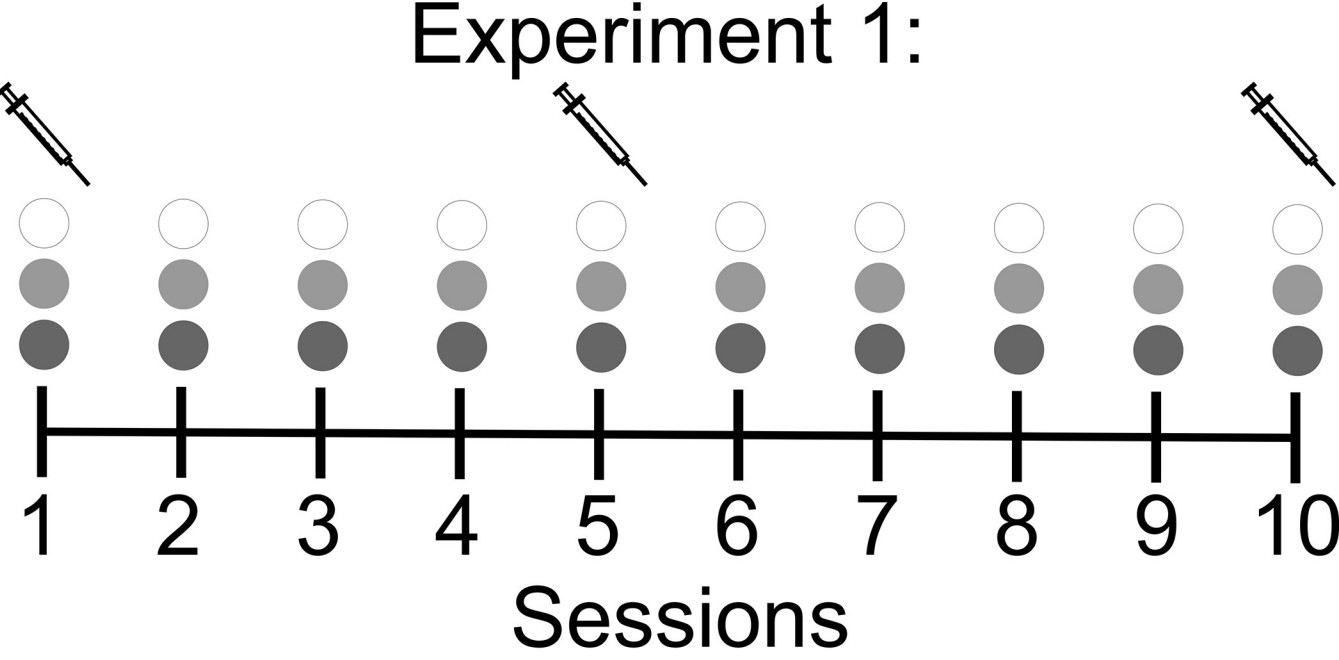

**Fig 1. Timeline of the behavioral tests to evaluate the effects of manganese administration.** Subjects were tested once a week for 10 consecutive weeks for sexual behavior, immediately followed by running wheel and then by the rotarod test. Twenty-four hours before sessions 1, 5 and 10 subjects were injected with saline,8 or 16 mg/kg of MnCl₂.

session, the animal was placed in an empty cage and monitored until full recovery before they were returned to their home cage.

*MEMRI preprocessing.* Image preprocessing was performed with FSL (FMRI Software Library v6.0) and ANTs (Advanced Normalization Tools, v2.1) software libraries. We used the function *DenoiseImage* for denoising. Then, we created an average brain (as template) using the set of scans from session 1 Multivariate Template Construction tool. Once the template was created, spatial normalization was performed on denoised images, using the ANTs Registration SyN tool, and then the scans from sessions 1, 5 and 10, were registered to the template. Then, we use the tool *fslmerge* from FMRIB Software Library v6.0 to merge all the aligned scans. The images were checked for quality control to eliminate images with artifacts or bad coregistration. Signal intensity normalization was performed extracting the data from the Harderian gland and dividing each image with its own value using *fslmaths*. Finally, the BET (Brain Extraction Tool) was used to remove all non-brain tissue from the intensity-normalized images.

*Image analysis.* For the image analysis, regions of interest (ROIs) were drawn using the template and the Paxinos and Watson rat atlas as guide [36]. The ROIs were from olfactory bulb (OB), bed nucleus of the stria terminalis (BNST), medial preoptic area (MPOA), ventromedial

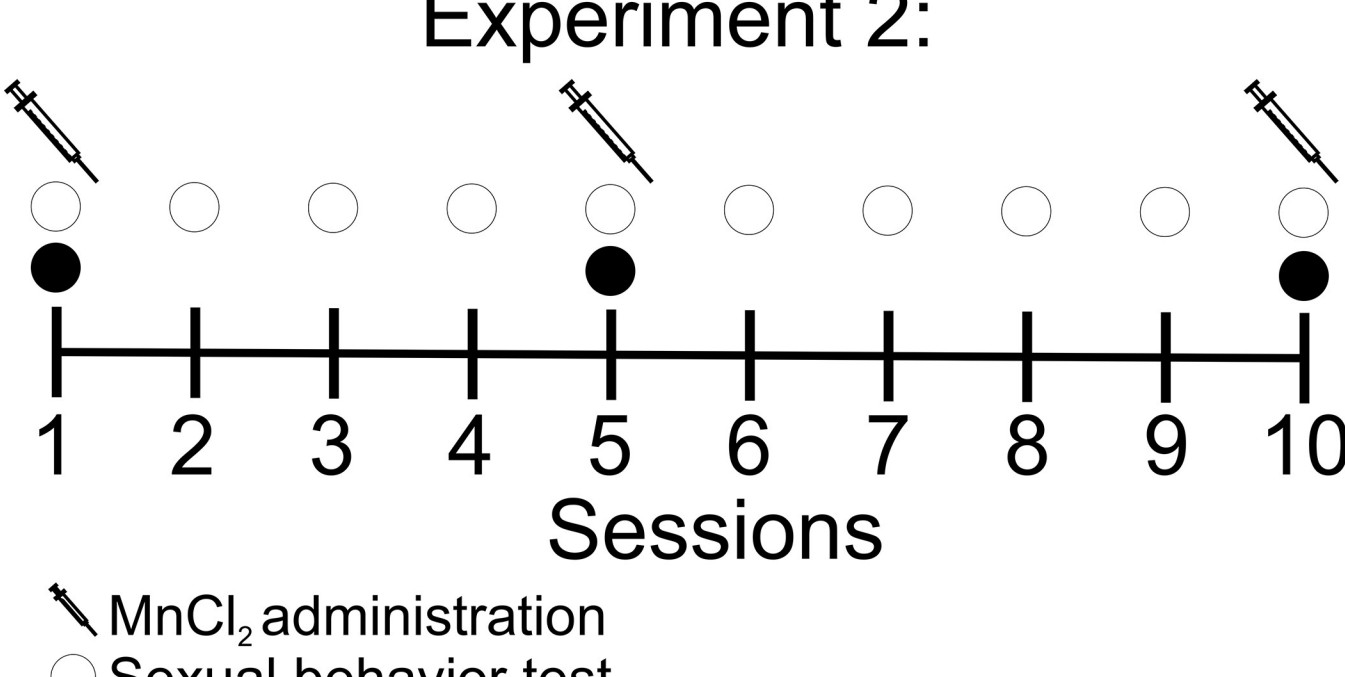

**Fig 2. The timeline of experiment 2 was similar to that of experiment 1.** Subjects were tested once a week for 10 consecutive weeks for sexual behavior. The control groups were left alone in the mating cage for 30 minutes. Twenty-four hours before sessions 1, 5 and 10 subjects were injected with 8 or 16 mg/kg of MnCl₂ depending on the group. They mated pacing the sexual interaction. The control females did not mate. Immediately after the test, MR images were obtained.

hypothalamus (VMH), nucleus accumbens (NAcc), amygdala (AMG), the hippocampus (Hipp), the striatum (STR) and the ventral tegmental area (VTA). For signal normalization, a ROI was created from the Harderian gland [37]. As a control region, a mask from the pituitary gland was created. The extraction of the average intensity from the ROIs was carried out using the FSLUTILS command-line *fslmeants*.

*Data analysis.* The behavioral and MRI data had no homogeneity of variance or normal distribution. In experiment 1, sexual behavior, running wheel and rotarod data were analyzed with a Friedman one Way ANOVA for repeated measures to compare within groups along the 10 sessions. Comparisons between groups in each session were done by a Kruskal Wallis (KW) test followed, in case of significance effects, by a Mann-Whitney U Test (MWU) as a *post hoc* test. In experiment 2, sexual behavior data was analyzed using the Mann-Whitney U Test to compare the two groups of females that mated.

The MRI data for each ROI was also analyzed by a Friedman one Way ANOVA for repeated measures to compare within groups in sessions 1, 5 and 10 and Tukey *post hoc* test in case of significant differences. Comparisons between groups with the same dose in the same sessions were done by the MWU test. The signal intensity of the OB, the BNST, the AMG, the MPOA and the VMH was grouped to analyze the socio-sexual circuit and the signal intensity of the NAcc, the STR, the Hipp and the VTA was grouped to analyze the reward circuit. The Freedman one way ANOVA was used to compare the activity of the circuits along the sessions

and comparisons between groups in the same sessions were done by a Kruskal Wallis test followed, in case of significant effects, by the MWU test.

A Cohen's D analysis was applied to the images. Briefly, we obtained the group mean from session 1, 5 and 10; the mean of session 1 was subtracted from the mean of session 10, and then divided with the standard deviation from session 1; the same procedure was applied with session 5, subtracting the mean of session 1 and dividing the result with the standard deviation of session 1. This was also done to compare sessions 5 and 10. Then, a voxel-wise analysis was performed on the spatial and signal intensity normalized images. Using the *Glm* tool from FSL, contrasts for the group comparisons were created, and the *randomise* function was used to perform the analysis. P-value corrected images were binarized using *fslmaths* and threshold to 0.95. Finally, the binarized images were corrected with the results from the Cohen's D analysis.

## Results

### Experiment 1

In the first experiment we wanted to evaluate if doses of 8 or 16 mg/kg of $MnCl_2$ altered sexual behavior, running wheel or the rotarod test.

**Sexual behavior test.** No significant differences were observed between groups on sessions 1, 5 and 10. As well no significant differences were observed between sessions for any of the sexual behavior parameters evaluated (see Table 1). The statistical values for each sexual behavior variable are presented on S1 Table.

**Running wheel test.** When we compared the running distance per session for the different groups (Fig 3) there were significant differences in sessions 6, 8 and 9 (Kruskal Wallis test, session 6, H = 5.985, df = 2, p = 0.05; session 8, H = 6.14, df = 2, p = 0.046; session 9, H = 10.905, df = 2, p = 0.004). In sessions 6 (MWU = 11, T = 89, p = 0.028) and 8 (MWU = 7,

**Table 1. Sexual behavior parameters in sessions 1, 5 and 10, in females treated with saline, 8 or 16 mg/kg of MnCl₂ 24 hours before testing with a sexually experienced male in experiment 1.** Data are expressed as mean ± SEM. (III: Inter intromission interval; MLI: Mean lordosis intensity; LQ: Lordosis quotient).

| Groups | Control (n = 9) | | | 8 mg/kg (n = 8) | | | 16 mg/kg (n = 9) | | |
|---|---|---|---|---|---|---|---|---|---|
| Sessions | 1 | 5 | 10 | 1 | 5 | 10 | 1 | 5 | 10 |
| Mounts | 12.67 ± 4.2 | 4.4 ± 1.4 | 6.5 ± 2 | 15.6 ± 6.7 | 5.9 ± 1.7 | 11 ± 5.3 | 10.3 ± 4.2 | 10.8 ± 2.7 | 2.8 ± 1 |
| Intromissions | 15.1 ± 1.5 | 11.7 ± 2.6 | 10.3 ± 2.7 | 16.4 ± 2.6 | 14.2 ± 2.5 | 4.5 ± 1.5 | 13.7 ± 2.4 | 14.2 ± 2.6 | 12.2 ± 5.5 |
| Ejaculations | 1.3 ± 0.4 | 1.4 ± 0.5 | 1 ± 0.4 | 1.5 ± 0.5 | 1.5 ± 0.4 | 0.5 ± 0.3 | 0.8 ± 0.3 | 1.4 ± 0.3 | 1.7 ± 0.4 |
| **Latencies** | | | | | | | | | |
| Mounts | 69.8 ± 22.8 | 239.4 ± 146 | 156.2 ± 64.3 | 55.9 ± 26.7 | 92 ± 41.7 | 304 ± 121.4 | 114 ± 56.6 | 264 ± 94.8 | 28.78 ± 10.7 |
| Intromissions | 150.1 ± 55.8 | 291.2 ± 175 | 239.5 ± 103 | 252.4 ± 113 | 278 ± 137.7 | 214.2 ± 100 | 187 ± 56.6 | 303 ± 97.6 | 78.9 ± 31.4 |
| Ejaculations | 390 ± 123.5 | 302.1 ± 110 | 505 ± 199.6 | 369.7 ± 123 | 530 ± 155.3 | 454.7 ± 223 | 426 ± 170 | 1029 ± 222 | 540 ± 151.5 |
| **III** | 39.4 ± 12.1 | 28.9 ± 9.8 | 36.2 ± 12.1 | 30.9 ± 11.8 | 44.9 ± 13.6 | 29.4 ± 14.8 | 26.2 ± 11 | 71.7 ± 17 | 43.6 ± 10.8 |
| **MLI** | 2 ± 0 | 1.8 ± 0.2 | 1.5 ± 0.3 | 1.97 ± 0.01 | 2 ± 0 | 1.5 ± 0.3 | 1.7 ± 0.2 | 1.99 ± 0 | 1.5 ± 0.3 |
| **LQ** | 100 ± 0 | 88.9 ± 11.1 | 77.8 ± 14.7 | 100 ± 0 | 100 ± 0 | 75 ± 16.4 | 88.9 ± 11 | 100 ± 0 | 77.8 ± 14.8 |
| **Return latencies after** | | | | | | | | | |
| Mounts | 9.6 ± 4.7 | 7.4 ± 3.9 | 4.7 ± 2.4 | 9.9 ± 3.7 | 9.9 ± 3.7 | 7.5 ± 3.7 | 11.1 ± 4.2 | 11.1 ± 4.2 | 5.6 ± 2.6 |
| Intromissions | 39.6 ± 3.9 | 29.2 ± 5.4 | 42.2 ± 17.3 | 46 ± 12.3 | 46 ± 12.3 | 24.3 ± 8.3 | 42 ± 7.8 | 42 ± 7.8 | 27.9 ± 6.2 |
| Ejaculations | 63.3 ± 21 | 48 ± 22.7 | 73.2 ± 33.6 | 69.6 ± 28 | 69.6 ± 28 | 45 ± 23.4 | 58.9 ± 37.1 | 58.8 ± 37 | 118.3 ± 41.8 |
| **Percentage of exits after** | | | | | | | | | |
| Mounts | 18 ± 6.5 | 23.1 ± 12.9 | 9.9 ± 5.8 | 22.6 ± 12 | 34 ± 12.8 | 7.6 ± 4.6 | 10.4 ± 6.5 | 16.3 ± 3.9 | 28.7 ± 12.8 |
| Intromissions | 71.5 ± 7.8 | 71.9 ± 10.5 | 70.6 ± 13.8 | 71.3 ± 7.5 | 81.7 ± 4.3 | 51.6 ± 15.7 | 45.2 ± 10 | 83.6 ± 3.3 | 66.6 ± 13.6 |
| Ejaculations | 100± 0 | 100 ± 0 | 100 ± 0 | 100± 0 | 100 ±0 | 100 ±0 | 100 ±0 | 1000 ±0 | 100 ±0 |

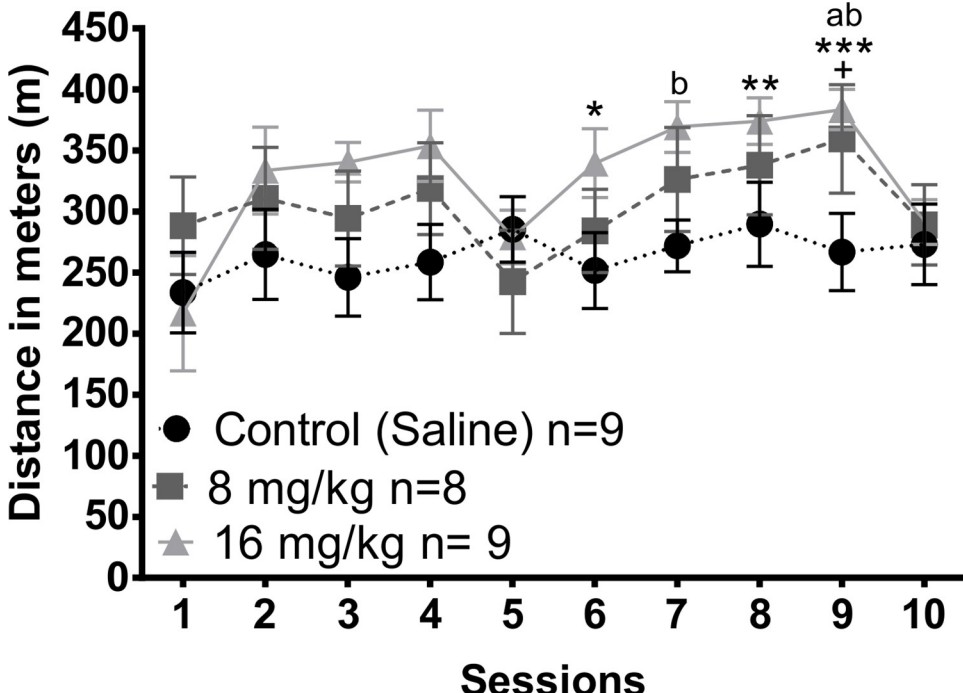

**Fig 3. Distance traveled in the running wheel test in the different group of animals along the 10 weeks of testing.**
Twenty-four hours before sessions 1, 5 and 10 subjects were injected with saline, 8 mg/kg or 16 mg/kg of $MnCl_2$. Data presented as mean +- SEM. + Difference between 8 mg/kg and control group, p<0.05. * Difference between 16mg/kg and control group, p<0.05; **, p<0.01; ***, p<0.001. a Difference from session 5 within the 8 mg/kg, p<0.05. b Difference from session 1 within the 16 mg/kg, p<0.05.

T = 93, p = 0.007) animals treated with 16 mg/kg ran a longer distance compared to the control group. In session 9 subjects treated with 8 mg/kg (MWU = 13, T = 49, p = 0.05) and 16 mg/kg (MWU = 2, T = 38, p = <0.001) ran a longer distance compared to the control group. When we compared the running distance for each group along the different sessions using the Friedman repeated measures ANOVA, significant differences were found in the groups injected with 8 ($X^2$ = 17.344, df = 9, p = 0.044) and 16 mg/kg ($X^2$ = 26.55, df = 9, p = 0.002). Post hoc tests revealed that in the 8 mg/kg group subjects ran longer distance in sessions 9 in comparison to session 5 (q = 4.613). In the group treated with 16 mg/kg subjects ran a longer distance on session 7 (q = 4.554) and 9 (q = 4.554) in comparison to session 1. No differences were found in the control group along the 10 sessions ($X^2$ = 9.335, df = 9, p = 0.407). The statistical values for the different running wheel comparisons are presented on S2A and S2B Table.

**Rotarod test.**    After the running wheel test, subjects were taken to the rotarod apparatus and were evaluated in two modalities, constant velocity (10 rpm) and increased velocity (10–15 rmp) with a maximum duration of 3 min for each test.

**Constant velocity.**    According to the Kruskal Wallis test, there were no significant differences in the fall latency in the constant velocity test (10 rpm) between groups along the 10 weeks of testing (Fig 4). As well, the Friedman repeated measures ANOVA revealed no significant differences along the sessions within each group. The statistical values are presented on S2A and S2B Table.

**Increased velocity.**    No differences were found in the fall latency in the increased velocity (10–15 rpm) test in the 10 weeks of testing (Fig 5). No differences between groups along the

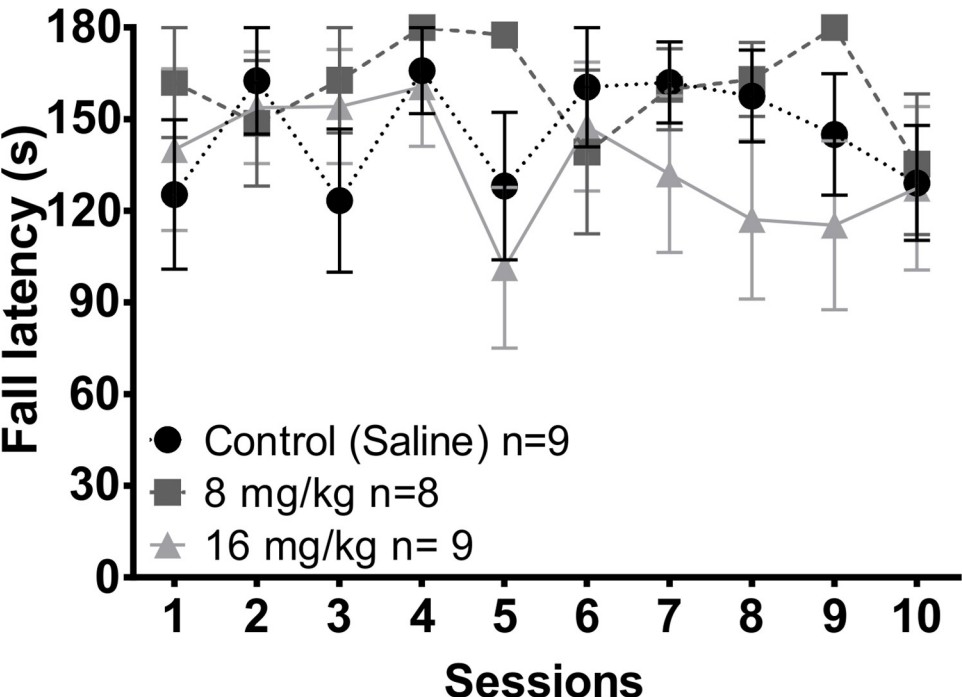

**Fig 4. Fall latency in the constant velocity (10 rpm) rotarod test in the different groups of animals along the 10 weeks of testing.** Twenty-four hours before sessions 1, 5 and 10 subjects were injected with saline, 8 mg/kg or 16 mg/kg of $MnCl_2$. Data are presented as mean +- SEM.

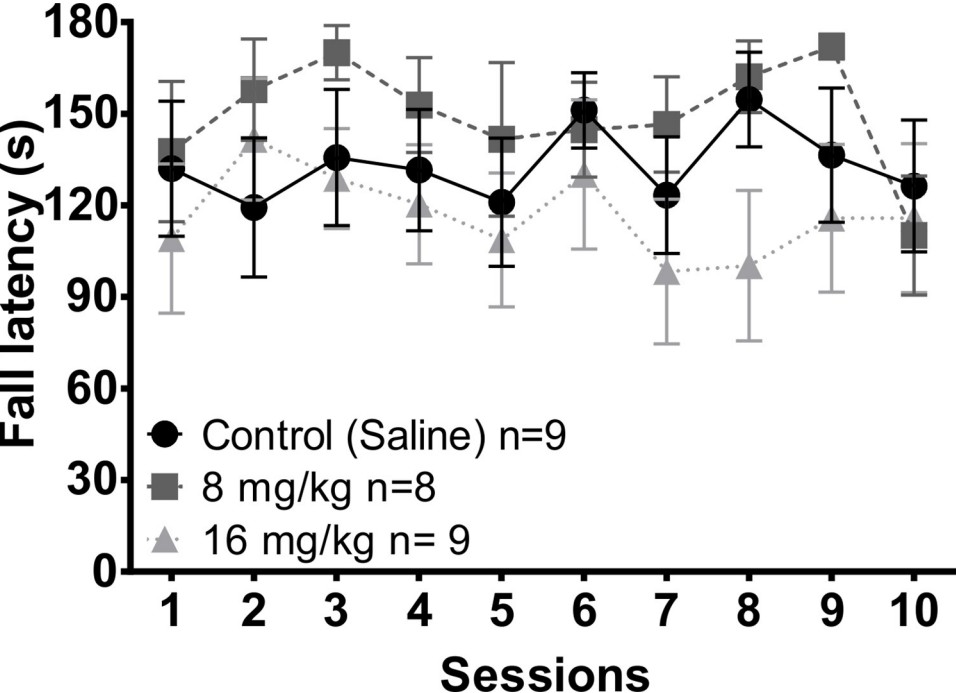

**Fig 5. Fall latency in the increased velocity (10–15 rpm) rotarod test in the different groups of animals along the 10 weeks of testing.** Twenty-four hours before sessions 1, 5 and 10 subjects were injected with saline, 8 mg/kg or 16 mg/kg of $MnCl_2$. Data are presented as mean +- SEM.

sessions were found based on the Kruskal Wallis test. The statistical values are presented on the S2A and S2B Table.

Together, the results of experiment 1 indicate that the doses of 8 and 16 mg/kg of $MnCl_2$ do not induce behavioral alterations that could interfere with the interpretation of the MRI results.

## Experiment 2

As already described, to evaluate the signal intensity induced by $MnCl_2$ four additional groups were used: females that did not mated injected with 8 or 16 mg/kg of $MnCl_2$ and females that mated pacing the sexual interaction injected with 8 or 16 mg/kg of $MnCl_2$. Females were tested once a week for 10 weeks and taken to the MRI facility and scanned on weeks 1, 5 and 10.

**Sexual behavior test.** Results of session 1, 5 and 10 are presented in Table 2. There was a significant difference in the mount latency in session 10 between groups (MWU = 38, T = 129, p = 0.018), with the 16 mg/kg group having a larger mount latency. As can be seen, the number of ejaculations received by the females was higher (Friedman test, $X^2 = 8.977$, df = 2, p = 0.011) in sessions 5 in comparison to session 1 (q = 3.467) in the 8 mg/kg group. Mount latency was reduced in session 10 in comparison to session 1 in the group treated with 8 mg/kg (FT, $X^2 = 6.157$, df = 2, p = 0.046; Tukey post hoc, q = 3.467). There was a significant increase in the percentage of exits after intromissions in both groups in sessions 5 and 10 with respect to their own session 1 (8 mg/kg group, $X^2 = 12.118$, df = 2, p = 0.002),session 5 vs session 1 (Tukey Test, q = 3.744), session 10 vs session 1 (Tukey Test, q = 4.576); the 16 mg/kg group ($X^2 = 7.385$, df = 2, p = 0.025), session 5 vs session 1 (TT, q = 3.328) and session 10 vs session 1 (TT, q = 3.328). The statistical values are presented on the S3 Table.

**Table 2. Sexual behavior parameters in sessions 1, 5 and 10, in females treated with 8 and 16 mg/kg of $MnCl_2$ 24 hours before testing and scanned.** Data are expressed as mean ± SEM. (III: Inter intromission interval; MLI: Mean lordosis intensity; LQ: Lordosis quotient).

| Groups | 8 mg/kg (n = 13) | | | 16 mg/kg (n = 13) | | |
|---|---|---|---|---|---|---|
| Sessions | 1 | 5 | 10 | 1 | 5 | 10 |
| Mounts | 13.3 ± 3.57 | 12.4 ± 2.72 | 4.46 ± 1.46 | 10.85 ± 2.71 | 10.35 ± 2.41 | 6.00 ± 1.71 |
| Intromissions | 12± 1.61 | 15.84 ± 2.1 | 15.69 ± 1.96 | 10.28 ± 1.95 | 14.92 ± 2.09 | 13.00 ± 2.19 |
| Ejaculations | 0.76 ± 0.25 | **1.76 ± 0.2[a]** | 1.69 ± 0.26 | 0.57 ± 0.20 | 1.42 ± 0.27 | 1.57 ± 0.30 |
| **Latencies (sec)** | | | | | | |
| Mount | 79.7 ± 25.6 | 59 ± 25.73 | **13.4 ± 4.0[a]** | 114.5 ± 27.3 | 288 ± 122.9 | **173.6 ± 89.3[*]** |
| Intromission | 129.23 ± 37 | 98.84 ± 56 | 26.69 ± 11 | 172.6 ± 46.5 | 249.9 ± 78.25 | 204.92 ± 88.4 |
| Ejaculation | 376.7± 125 | 630± 134.8 | 546± 128.7 | 411.78 ± 149 | 708.8 ± 158.3 | 587.8± 138.4 |
| **III (sec)** | 44.8 ± 18 | 57.1 ± 9.95 | 53.6 ± 13.18 | 40.9 ± 15.63 | 48.89 ± 9.41 | 53.32 ± 13.31 |
| **MLI** | 1.84 ± 0.15 | 1.82 ± 0.15 | 1.83 ± 0.15 | 1.71 ± 0.19 | 2.00 ± 00 | 1.85 ± 0.14 |
| **LQ** | 92.3 ± 7.69 | 92.3 ± 7.69 | 92.30 ± 7.69 | 85.71 ± 9.70 | 100.00 ± 00 | 92.85 ± 7.14 |
| **Return latency (sec)** | | | | | | |
| Mount | 23.8 ± 8.87 | 3.76 ± 2.05 | 3.59 ± 1.96 | 14.67 ± 6.54 | 4.88 ± 2.06 | 2.42 ± 1.27 |
| Intromission | 68.53 ± 21 | 34.86 ± 9.6 | 38.85 ± 15.5 | 41.5 ± 15.36 | 34.70 ± 5.91 | 44.02 ± 17.6 |
| Ejaculation | 23± 11.46 | 128 ± 42.36 | 102.8 ± 25.7 | 31.8 ± 19.57 | 94.7± 21.6 | 148.1± 66.2 |
| **Percentage of exits after** | | | | | | |
| Mount | 9.89 ± 5.02 | 8.54 ± 5.77 | 15.71 ± 6.74 | 9.77 ± 4.36 | 12.55 ± 5.33 | 11.22 ± 5.60 |
| Intromission | 33.91 ± 7.1 | **69.14± 7[a]** | **76.5± 7.15[a]** | 28.05 ± 8.34 | **70.4± 8.75[a]** | **74.69 ± 7.14[a]** |
| Ejaculation | 100±0 | 100 ±0 | 100± 0 | 100 ±0 | 100 ±0 | 100± 0 |

[*] Different from the 8 mg/kg in the same session, p<0.05

a Different from session 1 in theirs corresponding group, p<0.05

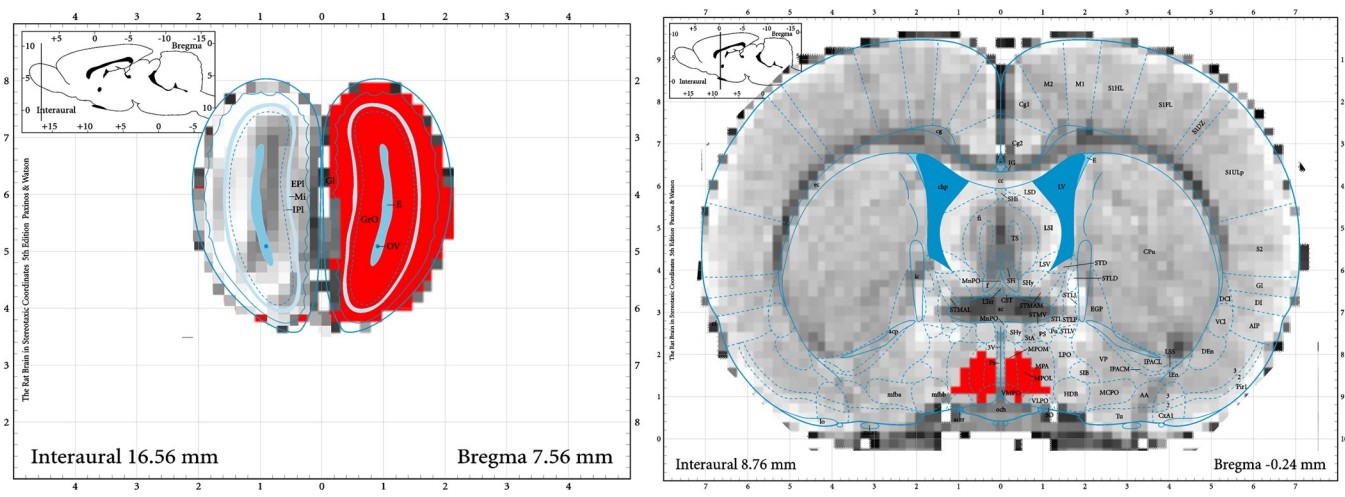

**Fig 6. Examples of ROIs using the atlas of Paxinos as reference [36].** From left to rigth, the olfactoy bulb and the medial preoptic area.

**MEMRI.** The images obtained were analyzed using regions of interest (ROI) from brain areas that control sexual behavior, examples of the *ROIs* used are presented in Fig 6. The four groups were compared on sessions 1, 5 and 10. With the 8 mg/kg dose we found significant differences between the group that mated and the control group, injected with $MnCl_2$. However, we found no differences with respect to session 1 (see S1 and S2 Figs). On the other hand, the 16 mg/kg dose clearly showed significant differences with respect to the control group and against session 1 indicating that this dose allows us to observe changes in signal intensity with behavioral experience. The statistical values for each comparison for each ROI are presented on S4A, S4B, and S5A Tables.

**Olfactory bulb.** A higher signal intensity (Fig 7) was observed in session 10 in the SB 16 mg/kg group (MWU = 27, T = 118, p = 0.003), in comparison to the control group. A significant difference was observed along the sessions in the SB 16 mg/kg group ($X^2$ = 10.308, df = 2, p = 0.006). Tukey *post hoc* analysis revealed a higher signal intensity in session 10 in comparison to session 1 (q = 4.438).

**Bed nucleus of the stria terminalis.** Results showed an increased signal intensity (Fig 7) in session 10 in comparison to session 1 (FT, $X^2$ = 10.308, df = 2, p = 0.006, Tukey *post hoc*, q = 4.438) in the SB 16 mg/kg group.

**Amygdala.** Significant differences were found (Fig 7) in session 10, with higher signal intensity observed in the SB 16 mg/kg group against its control (MWU = 40, T = 131, p = 0.024). The SB 16 mg/kg group showed an increased signal intensity in session 10($X^2$ = 12.423, df = 2, p = 0.002) in comparison to session 5 (Tukey *post hoc*, q = 3.328) and session 1 (Tukey *post hoc*, q = 4.992).

**Medial preoptic area.** In the MPOA (Fig 7) higher signal intensity was observed in the SB 16 mg/kg group in comparison to the control group in session 10 (MWU = 50, T = 131, p = 0.024). This group also showed a higher signal intensity in session 10 in comparison to session 1 ($X^2$ = 11.231, df = 2, p = 0.004, Tukey *post hoc*, q = 4.715).

**Ventromedial hypothalamus.** Again, we found a significant higher signal intensity (Fig 7) in session 10 for the SB 16 mg/kg group compared to the control (MWU = 31.5, T = 122.5, p = 0.007). The SB 16 mg/kg group showed an increased signal intensity (FT, $X^2$ = 14, df = 2, p = <0.001) in session 10 (Tukey *post hoc*, q = 5.27) in comparison to session 1.

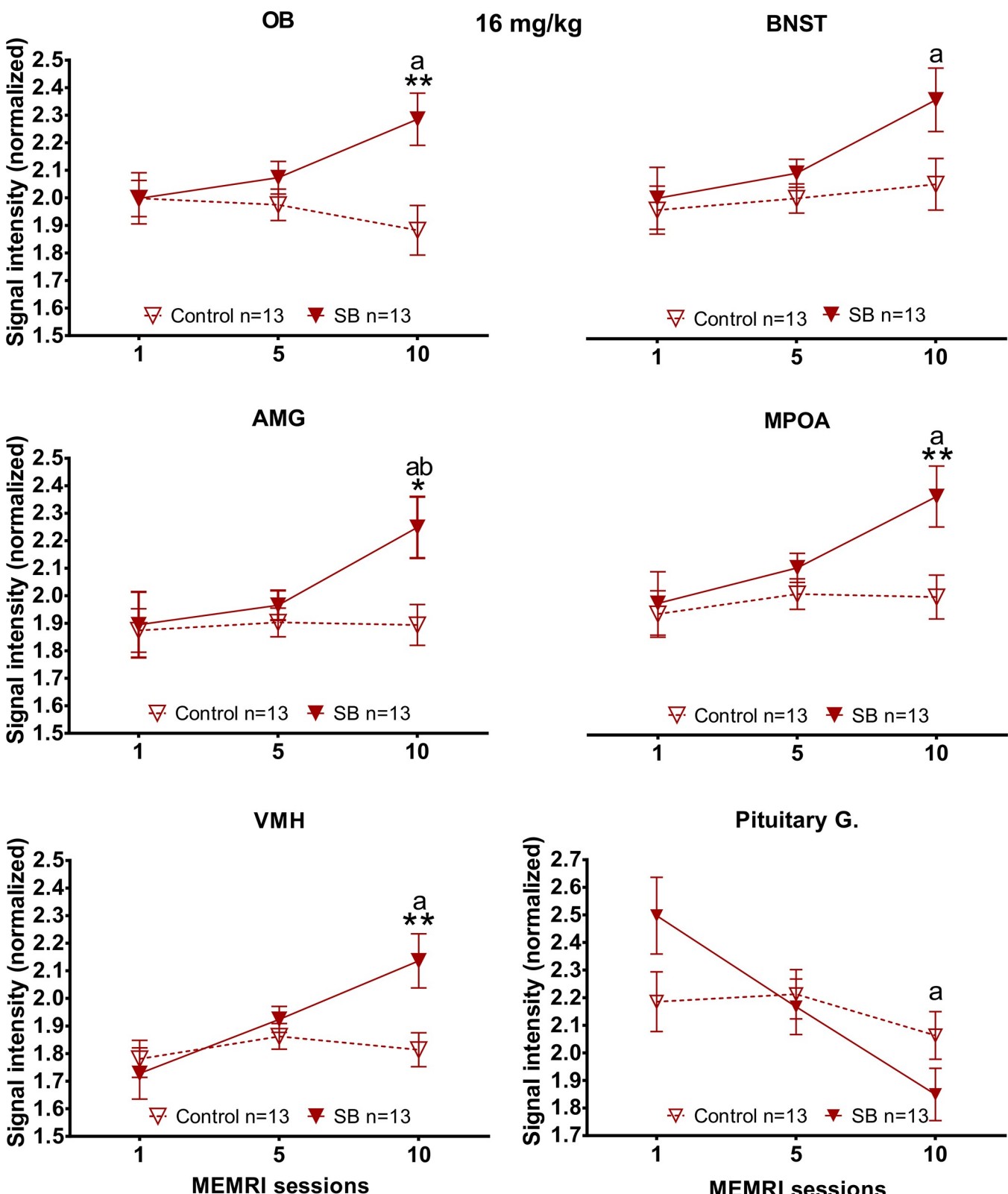

**Fig 7. Signal intensity of the olfactory bulb (OB), the bed nucleus of the stria terminalis (BNST), the amygdala (AMG), the medial preoptic area (MPOA), the ventromedial hypothalamus (VMH) and the pituitary gland in subjects treated with 16 mg/kg of MnCl₂.** Data are expressed as mean ± SEM. * Different from control p<0.05; ** p<0.01. a Different from session 1 p<0.05. b Different from session 5 p<0.05.

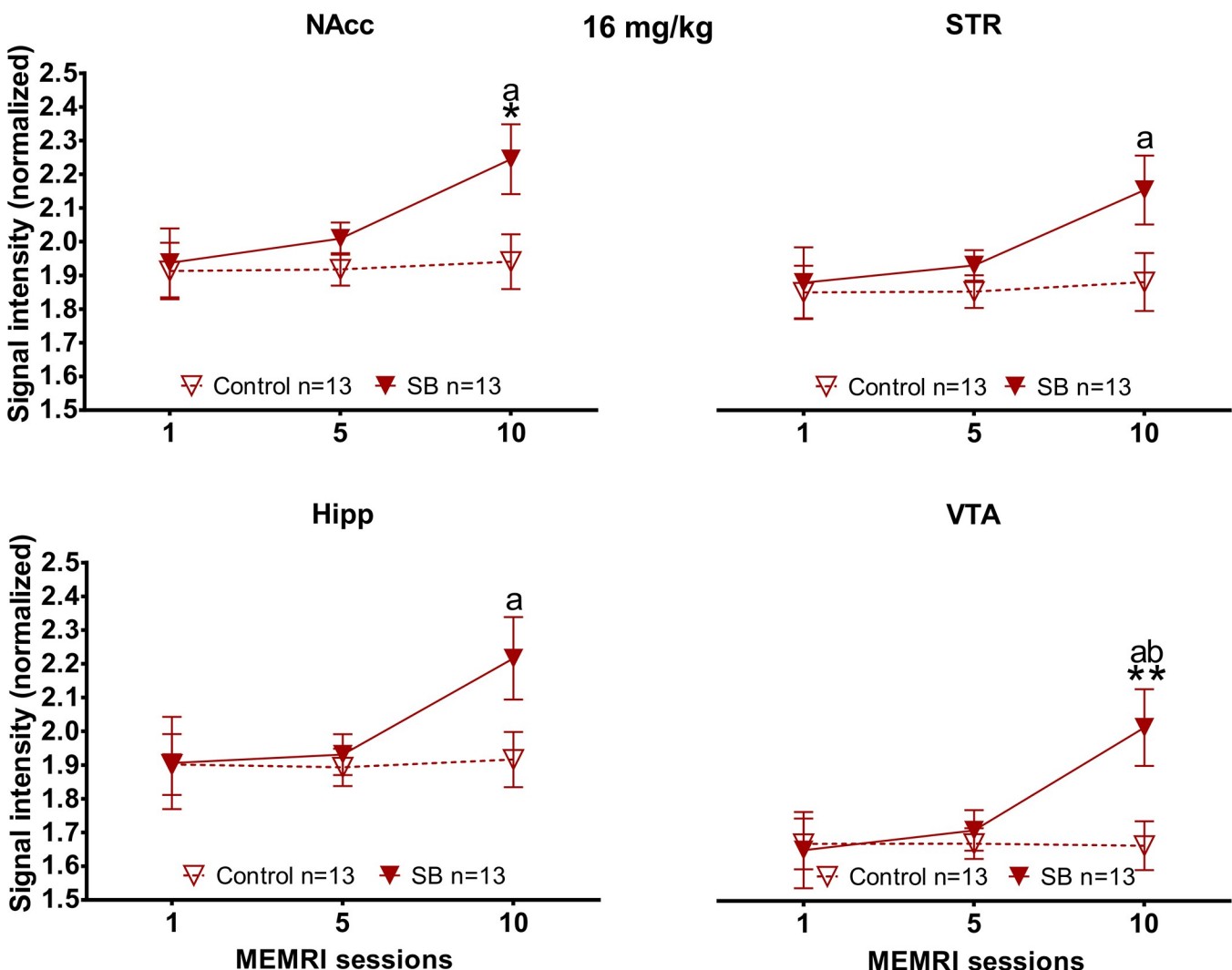

**Fig 8. Signal intensity in the nucleus accumbens (NAcc), the striatum (STR), the hippocampus (Hipp) and the ventral tegmental area (VTA) after the females paced the sexual interaction in subjects treated with 16 mg/kg of MnCl$_2$.** Data are expressed as mean ± SEM. * Different from control, $p < 0.05$; **, $p < 0.01$. a Different form session 1, $p < 0.05$. b Different form session 5, $p < 0.05$.

**Pituitary gland.** Signal intensity from the pituitary gland decreased in the SB 16 mg/kg group in session 10 (FT, $X^2 = 14$, df = 2, p = <0.001) compared to session 1 (Tukey *post hoc*, q = 5.27).

**Nucleus accumbens.** A significant increase in signal intensity (Fig 8) was observed in session 10 in the SB 16 mg/kg compared to its control (MWU = 44, T = 135, p = 0.04). The SB 16 mg/kg group also showed a higher signal intensity in session 10 ($X^2 = 9.846$, df = 2, p = 0.007) in comparison to session 1 (Tukey *post hoc*, q = 4.438).

**Striatum.** Significant differences were found in the SB 16 mg/kg group in session 10 against its session 1 (FT, $X^2 = 8$, df = 2, p = 0.018; TT, q = 3.883).

**The hippocampus.** Similar results were found in the hippocampus, with a higher signal intensity observed in session 10 ($X^2 = 9.294$, df = 2, p = 0.01) in comparison to its own session 1 (Tukey *post hoc*, q = 4.16) in the SB 16 mg/kg group.

**Ventral tegmental area.** Significant differences were observed (Fig 8) in session 10 between SB 16 mg/kg group versus its control (MWU = 33, T = 124, p = 0.009). Significant differences were found in the SB 16 mg/kg group along sessions ($X^2$ = 9.692, df = 2, p = 0.008), in session 10 a higher signal intensity was found in comparison to session 5 (Tukey *post hoc*, q = 3.328) and session 1 (Tukey *post hoc*, q = 4.16).

**Brain circuits.** We also analyzed the data comparing the average of the signal intensity of the areas associated with socio-sexual behaviors which included the OB, BNST, AMG, MPOA, and the VMH; and those associated with the reward circuit, NAcc, STR, Hipp, and VTA [38]. We analyzed the activation of both circuits in weeks 1, 5 and 10 with the dose of 16 mg/kg. The statistical values for the comparison for the brain circuits are presented on S5B and S5C Table.

As we can see in Fig 9, all groups had the same signal intensity in session 1 (KWT, H = 5.678, df = 3, p = 0.128). In session 5 the signal intensity was significantly increased (KWT, H = 28.326, df = 3, p = <0.001) in the socio-sexual circuit in the SB group compared to its control (MWU = 1559, T = 4811, p = 0.010), but not on the reward circuit (MWU = 1112.5, T = 2969, p = 0.12). In session 10 both circuits in the SB group had a significant increase in

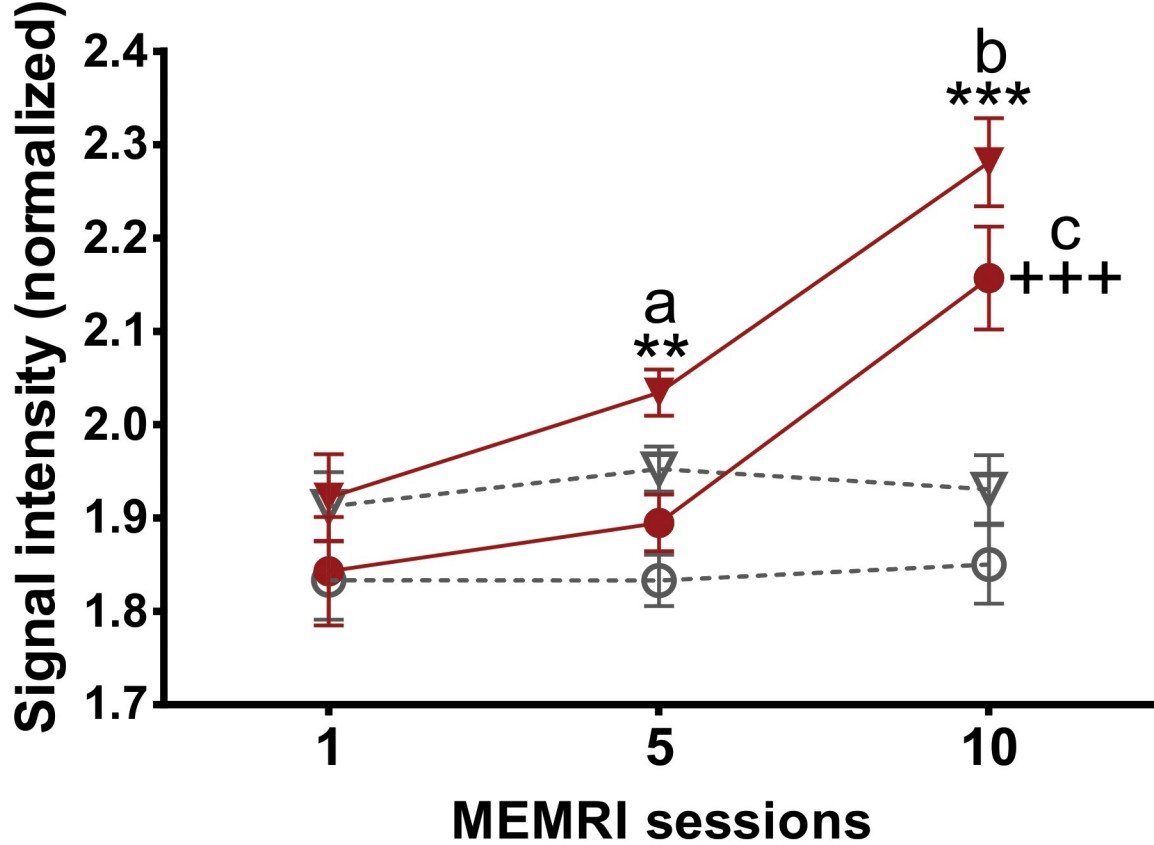

**Fig 9. Signal intensity in the socio sexual behavior and reward circuits in the control and in the group that paced the sexual interaction after treatment with 16 mg/kg of MnCl₂.** Data are expressed as mean ± SEM. ** Socio-sexual circuit, different from control and reward circuit in the same session p<0.01; ***, p<0.001. +++ Reward circuit different form its control, p<0.001. a Socio-sexual circuit, different from session 1, p<0.001. b Socio-sexual circuit, different from session1 and 5, p<0.01. c Reward circuit, different from session 1 and 5, p<0.001.

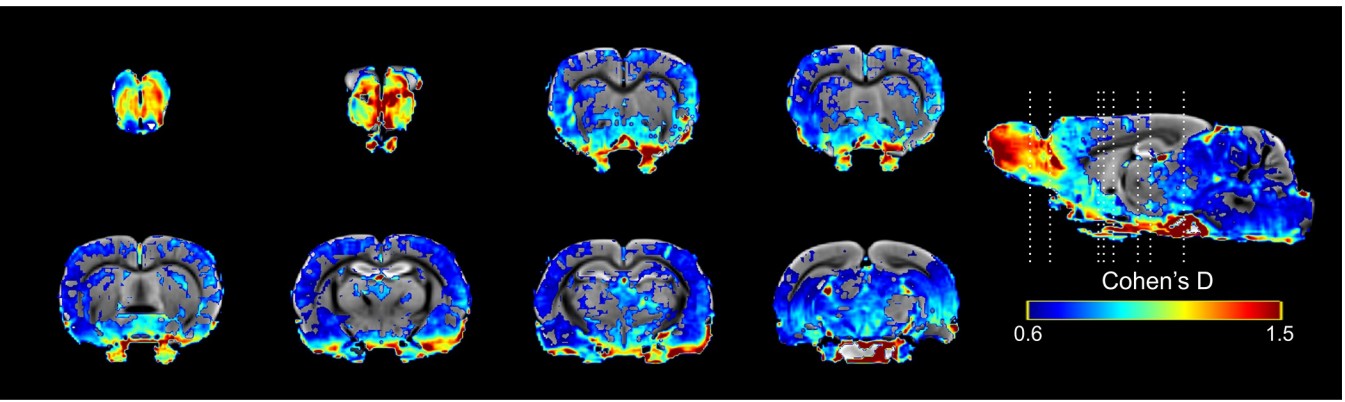

**Fig 10. Results from the Cohen's D analysis from the comparisons of control and sexual behavior from 16 mg/kg in session 10.**

signal intensity (KWT, H = 49.429, df = 3, p = <0.001) compared to their own control (Socio-sexual circuit, MWU = 946.5, T = 5423.5, p = <0.001; Reward circuit, MWU = 758.5, T = 3324, p = <0.001). Comparisons across time showed a significant increase of signal intensity in the Socio-sexual circuit in the SB group ($X^2$ = 57.723, df = 2, p = <0.001) in session 5 vs session 1 (TT, q = 4.217); and session 10 vs session 1 (TT, q = 10.667) and session 5 (TT, q = 6.45). Similar results were found in the Reward circuit in the SB group ($X^2$ = 36.203, df = 2, p = <0.001) in session 10 vs session 1 (TT, q = 8.321) and session 10 vs session 5 (TT, q = 5.616).

**Cohen's D analysis.** The Cohen's D analysis shows the changes in size effects between sessions and groups [39]. In our analysis, we compared the control group vs the sexual behavior group treated with 16 mg/kg in session 1, 5 and 10. No differences were found in session 1 and 5. In session 10, medium size changes (*d* = 0.6–0.8) were observed in different brain areas, but the larger changes in effect size (*d*>1) were found in the OB, the BNST, the MPOA, the VMH and the AMG (Fig 10). The NAcc, the STR and the VTA also showed large changes in effect size (*d* = 0.9). On the hippocampus, the main changes in size effect were observed in the dentate gyrus (*d*>1.2) more so than in the CA3 (*d* = 6) (Fig 10).

When we compared the signal intensity between sessions in the sexual behavior group (Fig 11), we found large effects in session 10 against session 1 in the OB, the BNST, the MPOA, the VMH and the AMG (d>1). In contrast, the effect size in the NAcc, the STR and the VTA were medium (d<0.8).

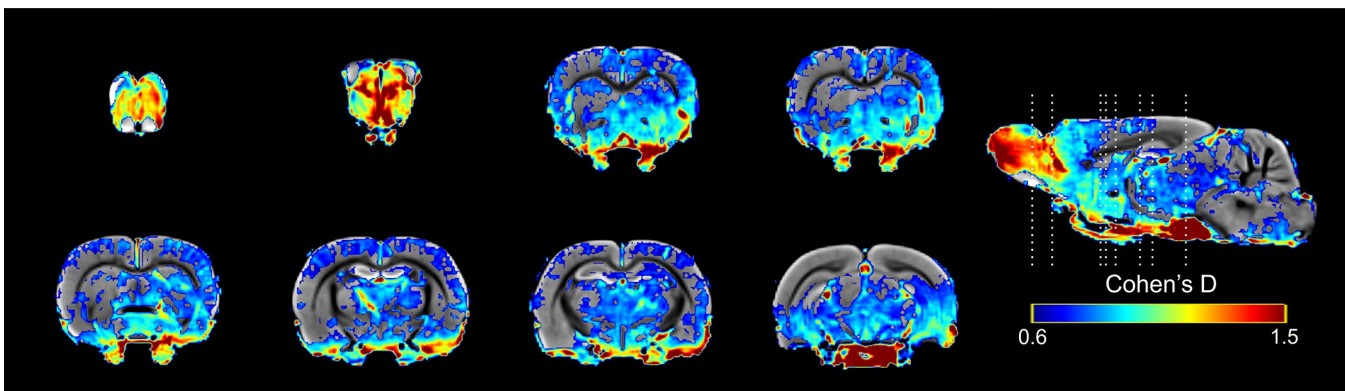

**Fig 11. Results of the Cohen's D analysis from the comparisons of sexual behavior group with 16 mg/kg in session 10 against its session 1.**

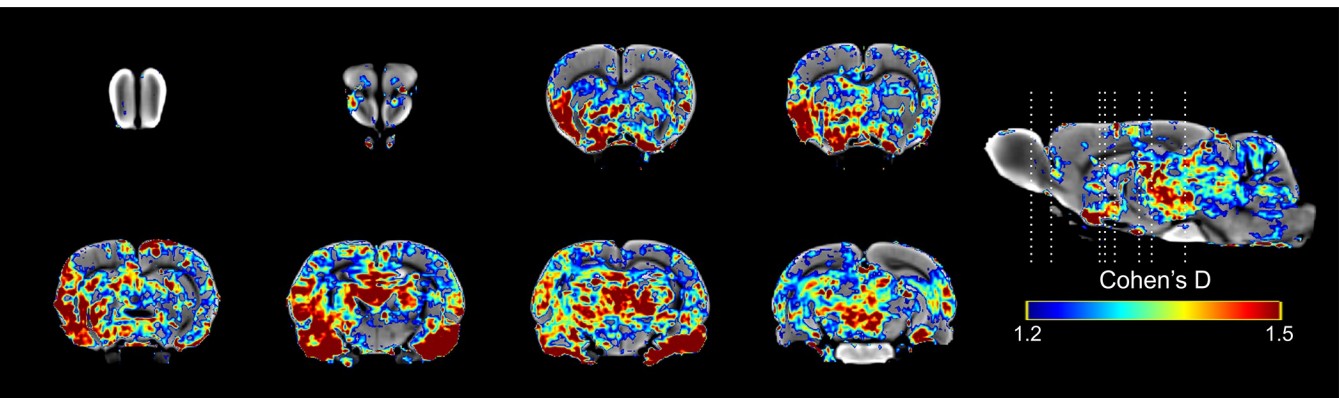

**Fig 12. Results of the Cohen's D analysis from the comparisons of sexual behavior group with 16 mg/kg in session 10 against its session 5.**

When comparing session 10 with session 5 (Fig 12) no further increase in effect size was observed in most of the areas studied.

## Discussion

The results of sexual behavior tests in experiment 1 and 2 clearly demonstrate that the administration of 8 or 16 mg/kg of MnCl$_2$ 24 hours before testing do not interfere with the display of sexual behavior of the female rat. All groups display similar levels of sexual receptivity as assessed by the lordosis quotient and the mean lordosis intensity. As well, we can infer that they were equally proceptive since no differences were found between the MnCl$_2$ treated groups and the control in the percentage of exits and the mean return latencies after receiving a mount, an intromission, or an ejaculation from the stimulus males. These measures of the paced mating condition are a better representation of sexual motivation than hops, darts and ear wiggling because the latter ones may or no may occur when females paced the sexual interaction [1] an in some cases they are reduced when females paced in comparison when they don't pace the sexual interaction [40]. It is also clear that the administration of MnCl$_2$ didn't modify any of the sexual parameters displayed by the females across the 10 weeks of testing. The MLI and LQ remained high, the percentage of exits and the return latencies are similar to what we [41–43] and others [1, 2] have consistently described.

The results of the RW test demonstrate that the administration of 8 or 16 mg/kg of MnCl$_2$ 24 hours before testing do not induce a reduction of RW on weeks 1, 5 and 10 demonstrating that these doses do not affect this motivated behavior. The distance traveled in sessions 6, 8 and 9 increased in comparison to the control group in animals treated with 16 mg/kg. These results could indicate that long term administration of MnCl$_2$ can increase locomotor activity. For example, it was shown that oral exposure to manganese in pre-weaning male rats (1–21 postnatal day) increased locomotor activity when tested at 24 days old [44]. However, in session 10 all groups ran the same distance making unlikely a long-term effect of MnCl$_2$ on voluntary activity. In a previous study it was shown that doses higher than 16 mg/kg produce a dose-dependent decrease in running wheel in male rats [35]. The administration of MnCl$_2$ was done 7 days apart and 3 hours before the running activity whereas in our study it was administered on weeks 1, 5 and 10, which is at least a 4-week interval between the injections and the running activity evaluated 24 hours after the injection.

No effects of MnCl$_2$ were observed in the forced rotarod test, neither in the constant velocity or the increased velocity test. The rotarod test evaluates fine motor coordination in rats and mice [45, 46]. Previous studies showed that the rotarod performance in adult male rats is

affected by administration, between postnatal days 8–12, of 20 mg/kg of $MnCl_2$ but no by 5 and 10 mg/kg [47]. In the studies described above where $MnCl_2$ had toxic effects the compound was administered in consecutive days while in our study there was at least a 4-week interval between the injections and the tests. Taken together, the behavioral effects of $MnCl_2$ on sexual behavior, running wheel and the rotarod indicate that the doses tested do not interfere with two motivated behaviors and does not induce nonspecific behavioral effects.

In our second study we evaluated the activation of different brain regions associated with sexual behavior along 10 weeks of testing. We tested the subjects with MEMRI on sessions 1, 5 and 10 to reduce the possibility of $MnCl_2$ overlapping. The control groups were injected with 8 or 16 mg of $MnCl_2$ but did not mate showing similar signal intensity in the three sessions indicating that manganese did not accumulate in our testing paradigm to induce an increase in the signal intensity. These observations indicate that the increase in signal intensity in the experimental groups is not caused by accumulation of manganese and overlapping of its administration but is due to the display of sexual behavior.

Our images analysis demonstrates that FLASH sequence is sensitive to a sc administration of 16 mg/kg of $MnCl_2$ allowing the detection of changes in brain activity across time within the same subject and in comparison, to a control group, potentiating the advantages of using MEMRI. The FLASH sequence is relatively short (25–45 minutes) with good contrast and resolution, making possible the identification of brain structures of rodents without losing anatomical accuracy. The use of the FLASH sequence is common in magnetic resonance protocols using manganese as contrast medium [48–51]. Uselman and colleagues detected the brain activity of free moving mice after a single fear exposure [49], and they compared the brain activity before and after the exposure to a predator odor, using 25 mM/$MnCl_2$. They identified increased neural activity in areas related to anxiety in the serotonin transporter knockout (SERT-KO) mice compared to WT. The data obtained using MEMRI was corroborated with c-Fos immunohistochemistry.

Our results agree with previous observations demonstrating that $MnCl_2$ can be used in rats to identify activation of different brain circuits without producing severe toxic effects affecting the behavior studied. A significant increase in brain activity was observed in several brain regions demonstrating that MEMRI can be used to map brain activity in an operant behavior in freely moving rats [30]. Similar results were observed in a delay matching to place water maze task. Although the subjects showed ataxia and erythema, that disappear 30 minutes after the administration of $MnCl_2$, they did not show alterations on memory performance or sensorimotor effects. The authors were able to study the enhancement of hippocampal signal without alterations of hippocampus dependent behavior using MEMRI [52]. MEMRI has also been used to identify central nervous activity in Locusts (grasshoppers). The experimental group was forced to walk on a treadmill for two hours while the control group was immobilized. A significantly higher signal was observed in the walking subjects in ganglions controlling walking behavior [27]. MEMRI has also shown its reliability for auditory brain mapping after a single injection in mice. Subjects with a unilateral conductive hearing loss at different ages were studied with MEMRI demonstrating the tonotopic organization of the mouse inferior colliculus [53]. These results demonstrate that MEMRI can be used, as a less invasive technique, to evaluate brain activity in longitudinal studies.

Neuronal activation after sexual behavior is well documented [10, 54–57]. Long-Evan female rats that received 14 mounts with intromissions including ejaculation expressed a higher number of FOS positive cells in the preoptic and amygdala brain regions compared to females that received 15 mounts without intromission, demonstrating that neuronal activation in these areas is related to more intense vaginocervical stimulation and not just cutaneous perineal stimulation [10]. Rowe and Erskine found that not only the preoptic and amygdala are

activated by intromissions, but also the BNST showed a higher FOS response compared to females that received only mounts [54]. The only mounts group had a significant increase of FOS positive cells in the ventromedial nucleus, paraventricular nucleus (PVN) and CTF compared with home cage females. The activation of these structures was related with the display of the lordosis reflex. Flanagan-Cato and McEwen found similar results using Fos along with Jun B and C expression in the preoptic area, the BNST, the PVN and the amygdala posterodorsal comparing females that mated with females that did not mate [55]. The increase in FOS expression appears to be related with the amount of stimulation that the females received. It has been observed that the intensity of the sexual stimulation received is associated with the increase in FOS expression. Females that received an ejaculation showed a higher FOS response than when they receive an intromission. As well, those that received intromissions showed a higher FOS response than those receiving mounts. This differential expression of FOS was observed in brain regions including the MePD, the MPOA, the BNST and the VHM [56]. The importance of these brain regions in the control of sexual behavior of females and males of several species, has been demonstrated by different techniques including among others, lesion, stimulation, anatomical, histological and, as already described, by expression of immediate early genes [58].

The use of early gene expression, like c-Fos, as an indirect marker of neuronal activity allows analyzing changes in brain activity after one mating session counting individual cells in the areas of interest. On the other hand, MEMRI requires the accumulation of manganese in enough neurons to change the local magnetic field. As expected, in session 1 no statistically significant differences were observed between groups. In the groups treated with 8 mg/kg a significant increase was observed in the subjects that mated in comparison to those that did not on session 5 in all areas studied except for the VTA. On session 10 similar results were observed, higher signal intensity was detected in the subjects that mated in all areas studied except for the AMG and Hipp. In the group treated with 16 mg/kg the subjects that mated showed a higher increase with respect to the subjects that did not mate in most areas studied except for the BNST, the STR and the Hipp. The activation of these brain regions is consistent with what has been shown with other techniques.

We also analyzed two important circuits that control behaviors in different species [38]. The socio-sexual network, which regulates parental care, aggression and other social behaviors including sexual behavior. The mesolimbic reward system reinforces responses to incentive stimuli, for example sex with a conspecific. The analysis of the brain circuit controlling socio-sexual behaviors as well as the reward circuit is another advantage of MRI. Both circuits are activated but the reward circuit showed a reduced activation in session 10 in comparison to the socio-sexual circuit. Different lines of evidence have shown that sexual behavior induces a reward state in both male and female rats which is mediated by opioids [3]. From our study it is clear that both circuits are activated by sexual behavior, but the sexual circuit is activated sooner and with a higher intensity than the reward circuit. Only future studies could determine if the socio-sexual circuit maintains higher signal intensity than the reward circuit with more sexual experience.

Contrary to our expectations the 16 mg/kg dose of $MnCl_2$ did not induce higher signal intensity compared to the 8 mg/kg dose on session 5. The differences became evident on session 10. The reason why the signal intensity in both 8 mg/kg groups decreased is unknown. Most studies use different manganese doses and administration routes. Brunnquell and colleagues reported the effect of three doses: 30, 45 and 60 mg/kg, with 3 animals per dose [59]. $R_1$ relaxation maps shows that doubling the dose does not increase significantly the median $R_1$. The controls groups treated with 8 or 16 mg/kg that did not mate showed similar signal intensity in session 1, 5 and 10 indicating that the differences observed in the SB groups are due to

the display of the behavior and not to the overlapping of manganese administrated doses. When the data of the *ROIs* were analyzed, a few differences were found when comparing session 5 with session 1. The sexual behavior test in the present experiment lasted 30 minutes, future experiments need to analyze if longer sessions, 60 minutes for example, can increase brain activity due to increase behavior resulting in more manganese accumulation and signal intensity. It was until session 10 that differences were evident, future studies need to determine if the increase in signal intensity can be observed between session 5 and 9.

To the best of our knowledge this is the first study using MRI, to evaluate longitudinal changes in brain activity in animals that repeatedly mated to determine the possible role of sexual experience upon the activity of brain circuits controlling sexual behavior. There are indeed few studies that have combined MRI and sexual behavior. One of those studies evaluated BOLD signal response to cocaine in females displaying or not sexual receptivity, as an indirect measure of hormone levels. Female rats that failed to display lordosis showed a greater activation of mesolimbic and nigrostriatal structures than females displaying lordosis in response to manual stimulation of the flanks. The receptive females also showed a lower response to cocaine than non-receptive females [60]. There are two studies in males in which MRI was used to assess whether neurotoxic lesions could be monitored by MRI and if the possible changes could be correlated with the behavioral alterations. In one study 6-hydroxydopamine (6-OHDA) was administered in the MPOA to reduce catecholamine (CA) neurons and fibers. The authors found a destruction of CA fibers in the MPOA by hyperintense MRI but the temporary alterations in sexual behavior were not related to changes in MRI [61]. In another study the same group evaluated the effect of NMDA lesions of the MPOA in sleep, temperature, and sexual behavior. They found that starting 3 hours after NMDA injection the brightness of the MPOA increased in the following 2 days. Again, the reduction of sexual behavior did not correlate with changes in MRI [62].

The Cohen's D analysis on the MERMI results shows that areas associated with sexual behavior, the OB, the BNST, the MPOA, the VMH and the AMG had a larger size effect. On the other hand, a medium size effect was observed in the NAcc, the STR, the Hipp and the VTA, suggesting that in order to induce a reward state, the socio-sexual circuit needs to be activated first by the increase of sexual experience.

## Conclusions

The administration of 8 or 16 mg/kg of MnCl$_2$ one month apart does not produce alterations in two motivated behaviors, sexual behavior and running wheel, and does not affect the rotarod test indicating that with this administration interval and doses we can evaluate changes in signal activity associated with a particular behavior without inducing unspecific effects. In experiment two we demonstrated that the sexual circuit is activated first and with higher intensity than the reward circuit. Our study demonstrates that MEMRI methodology using a 16 mg/kg dose of chloride manganese is optimal to use in longitudinal studies when multiple scans are required to evaluate changes in brain regions and circuits activated by motivated behaviors and how they might change with behavioral experience. The MEMRI methodology can be use in different species to evaluate in the same animal brain changes associated with a particular behavior, but possible toxic effects should be considered. It is also clear, that we can now evaluate many different aspects of sexual behavior and other motivated behaviors in both males and females, by MEMRI.

## Supporting information

**S1 Fig. Signal intensity of the olfactory bulb (OB), the bed nucleus of the stria terminalis (BNST), the amygdala (AMG), the medial preoptic area (MPOA), the ventromedial**

**hypothalamus (VMH) and the pituitary gland in subjects treated with 8 mg/kg of MnCl$_2$.** Data are expressed as mean ± SEM. * Different from control $p < 0.05$; ** $p < 0.01$; *** $p < 0.0001$.
(TIF)

**S2 Fig. Signal intensity in the nucleus accumbens (NAcc), the striatum (STR), the hippocampus (Hipp) and the ventral tegmental area (VTA) after the females paced the sexual interaction in subjects treated with 8 mg/kg of MnCl$_2$.** Data are expressed as mean ± SEM. * Different from control, $p < 0.05$; **, $p < 0.01$.
(TIF)

**S3 Fig. Signal intensity in the socio sexual behavior and reward circuits in the control and in the group that paced the sexual interaction after treatment with 8 mg/kg of MnCl$_2$.** Data are expressed as mean ± SEM. * Socio-sexual circuit, different from control in the same session $p < 0.05$; **, $p < 0.01$; ***, $p < 0.001$. ++ Reward circuit different form control in the same session, $p < 0.01$; +++ $p < 0.001$.
(TIF)

**S1 Table. Statistical results using the Kruskal Wallis and the Friedman repeated measures test on the sexual behavior parameters comparing the different groups in experiment 1.**
(DOCX)

**S2 Table.** a) Statistical results of the Kruskal Wallis test for the running wheel and rotarod (in both modalities) comparing the different groups in experiment 1.b). Statistical results comparing the running wheel and rotarod (in both modalities) using the Friedman repeated measures test in experiment 1.
(DOCX)

**S3 Table. Statistical results of the Mann Whitney and the Friedman repeated measures ANOVA test on parameters comparing the different groups in experiment 2.**
(DOCX)

**S4 Table.** a) Statistical results comparing the signal intensity of the different ROIs from 8 mg/kg group in experiment 2 using the Mann-Whitney U test. b) Statistical results comparing the signal intensity of the different ROIs from 16 mg/kg group in experiment 2 using the Mann-Whitney U test.
(DOCX)

**S5 Table.** a) Statistical results using the Friedman repeated measures ANOVA comparing the signal intensity in the different ROIs activated by sexual behavior in experiment 2. b). Results from the comparisons within groups using the Kruskal Wallis test on ROIs that activates sexual behavior from experiment 2. c). Statistical results comparing by Friedman repeated measures ANOVA the circuits activated by sexual behavior in experiment 2.
(DOCX)

## Acknowledgments

We thank Leopoldo Gonzalez, Omar González, Ramón Martínez, Martín García, Laura Sánchez and Sandra Hernandez for their excellent technical assistance; Alejandra Castilla León and María A. Carbajo for their support on providing the experimental subjects; Deisy Gasca for the assistance on the rotarod and running wheel tests and Francisco Javier Camacho for the support on the OVX and sexual behavior tests. Alejandro Aguilar-Moreno is a doctoral student from Programa de Doctorado en Ciencias Biomédicas, Universidad Nacional Autónoma de México (UNAM) and received fellowship 766439 from CONACYT.

## Author Contributions

**Conceptualization:** Luis Concha, Raúl G. Paredes.

**Formal analysis:** Juan Ortiz, Sarael Alcauter, Raúl G. Paredes.

**Funding acquisition:** Raúl G. Paredes.

**Investigation:** Alejandro Aguilar-Moreno, Juan Ortiz.

**Methodology:** Alejandro Aguilar-Moreno, Juan Ortiz, Luis Concha, Sarael Alcauter.

**Project administration:** Raúl G. Paredes.

**Resources:** Raúl G. Paredes.

**Supervision:** Raúl G. Paredes.

**Writing – original draft:** Alejandro Aguilar-Moreno.

**Writing – review & editing:** Luis Concha, Sarael Alcauter, Raúl G. Paredes.

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
