## [Decision Letter · Decision Letter 0]

30 May 2022

PONE-D-22-12289Brain circuits activated by female sexual behavior evaluated by manganese enhanced magnetic resonance imaging PLOS ONE

Dear Dr. Paredes,

Thank you for submitting your manuscript to PLOS ONE. After careful consideration, we feel that it has merit but does not fully meet PLOS ONE’s publication criteria as it currently stands. Therefore, we invite you to submit a revised version of the manuscript that addresses the points raised during the review process. Both reviewers believed this study is of great interest. However, both also provided several insightful suggestions related to interpretation and several required clarifications. Please take into consideration the comments provided by reviewers and include a point-by-point response to their suggestions in resubmission.

We look forward to receiving your revised manuscript.

Kind regards,

Juan M Dominguez, PhD

Academic Editor

PLOS ONE

Journal Requirements:

"This research was supported by grant DGAPA, PAPIIT, UNAM, IN206521. "

"This research was supported by grant DGAPA, PAPIIT, UNAM, IN206521

3. Please ensure that you refer to Figures 6 and 11 in your text as, if accepted, production will need this reference to link the reader to the figure.

Reviewers' comments:

Reviewer's Responses to Questions

**Comments to the Author**

1. Is the manuscript technically sound, and do the data support the conclusions?

Reviewer #1: Yes

Reviewer #2: Yes

2. Has the statistical analysis been performed appropriately and rigorously? 

Reviewer #1: Yes

Reviewer #2: Yes

3. Have the authors made all data underlying the findings in their manuscript fully available?

Reviewer #1: Yes

Reviewer #2: Yes

4. Is the manuscript presented in an intelligible fashion and written in standard English?

Reviewer #1: Yes

Reviewer #2: Yes

5. Review Comments to the Author

Reviewer #1: This study explored brain changes associated to sexual behavior in female rats via manganese enhanced MRI. The authors explored 3 different doses of MnCl2 (8, 16, 32 mg/kg). They refer that 32 mg/kg induced severe skin lesions, so they continued exclusively with 8 and 16 mg/kg, with no effects in skin, locomotion and sexual behavior. Then, they tested brain activity evoked by sexual behavior via MnCl2-enhaced MRI. They compared sessions 1, 5 and 10. Concluded that this approach allows identification of changes in brain circuits.

The manuscript is well written, interesting and shows novel data using sophisticated imaging analysis. Minor modifications/explanations are recommended:

In the last line of the abstract they conclude that "The socio sexual circuit is activated sooner and with higher intensity than the reward circuit". Such statement is not easy to understand in an abstract. Please expand to clarify the meaning of the sentence. For instance, in any given sexual encounter it is expected the socio sexual circuit to be activated before animals experience reward. So, what is the novelty about the sentence?

In experiment 2, authors did not evaluate social behavior, nor reward. Their analyses of sexual behavior included mainly male sexual behaviors (mounts, intromission, ejaculation), lordosis and return latencies. Authors did not evaluate proceptive behaviors either. So, we don´t know whether MnCl2 enhanced proceptivity at the long-run, or if enhanced MRI signal was consequence of more proceptivity. Please discuss.

In the methods section, they must report the dose of ketamine and xilacine in mg/kg (not as part of drug in ml)

Page 5, in experiment 1. It is not clearly explained they used the subcutaneous way to deliver the MnCl2. It is explained much later.

Figure legends and axes are blurry and difficult to read.

please discuss about the use of MnCl2 for MRI in other species, considering the potential side effects and welfare.

Reviewer #2: The paper by Aguilar-Moreno et al. explores which brain regions are activated following paced sexual experiences in female rats, using a fairly unique technique, manganese-enhanced MRI, or MEMRI. MEMRI presents the advantage of visualizing brain activity changes in living rats, as opposed to using more standard techniques such as immediate early gene expression in excised tissue. The goals of the study were to 1) establish a dose of MnCl2 that following administration did not cause changes in motivated behaviors such as sexual behavior, wheel running and rotarod walking and 2) quantify brain activation in motivation and reward circuits following paced mating.

This manuscript nicely provides a description of the pathway to using an uncommon procedure for assessment of brain activity in a rodent model of behavior. In this paper, the authors begin by laying out the procedures, including choice of doses of MnCl2 and behavioral assessments used to ensure safe dosing that would not interfere with the behavior of interest—paced sexual mating. MEMRI, as demonstrated in the current study, proves to be a useful tool that not only supports previous findings of investigations of brain activation in the socio-sexual and reward circuits using techniques such as FOS labeling, but also allows for real-time, repeated measures of neural activation. The authors found that neither doses of 8 nor 16 mg/kg of MnCl2 interferes with sexual performance or mobility. In addition, in comparison to female rats that did not mate, paced mating activated many regions in the socio-sexual and reward circuits, including the olfactory bulb, amygdala, VMH, VTA and nucleus accumbens.

While I do believe this manuscript presents important information and therefore approve of its publication, there are several points which need to be addressed prior to publication. Some of the procedures need further clarification in the Methods section. Additionally, there are points in which the results are difficult to follow because of inconsistencies in referencing tables or presentation of information within the figures. Furthermore, in the Discussion section, I believe the authors could further “sell” the importance of their findings and methodology by placing the study in a broader context of its usefulness, or how this study presents a foundation for new research. Please see below for specific recommendations for revision.

Recommended Changes:

General

• There are several instances throughout the paper that would benefit from review to address issues of grammatical clarity and concision.

Introduction

• It might be helpful for readers unfamiliar with MEMRI or MRI in general to explain the importance of “increasing MR contrast in T1 weighted images” (2nd paragraph, page 4).

• Emphasize in aim 2, at the bottom of page 4, that the study will investigate paced female sexual behavior, as brain activation patterns may differ between paced and non-paced mating.

Methods

• Paragraph 1, page 5 indicated that rats were primed with 25µg EB. This dose seems excessively high for induction of behavioral estrus. Could this be a typing mistake? If not, perhaps include a reference for another study in which this supraphysiological dose is used.

• It might be useful to describe how subject numbers were determined for Experiment 1, as the numbers seem low, and thus there is a good deal of variability in some of the behavioral measurements in the results section.

• While return latencies are adequate measures of sexual motivation, it would have been further strengthened had the authors also included other measures of proceptivity such as hopping and darting and ear-wiggling as mentioned in the introduction. Perhaps here, or elsewhere, the authors could provide some explanation as to why these behaviors were not also quantified, or why return latencies may provide comparatively better assessments of motivation.

• It may be useful to describe in further detail how percent exits and return latencies are calculated to improve clarity for readers unfamiliar with typical measures of paced mating behavior.

• The abstract states, “In experiment 1 we evaluated the effects of two doses of MnCl2, 8 and 16 mg/kg, upon female sexual behavior, running wheel and the rotarod once a week for 10 weeks.” This implies that in Exp 1, before each of the 10 tests, they were treated with MnCl2. However, in the Methods section on page 7, it is stated that in both experiments, manganese treatment only occurred prior to tests 1, 5, and 10. Please clarify either in the Methods or Abstract how many doses were administered in each experiment.

• Figure 1 X-axis should be labeled “Sessions” as in Figure 2.

Results

• Reporting of Experiment 1 results contains inaccurate references to supplementary tables. On page 11, it states, “The statistical values for each sexual behavior variable are presented on supplementary tables 1 and 2”. However, these data are only presented on supplementary table 1. Supplementary Table 2 contains statistics for the running wheel and rotarod tests.

• There is no reference to either supplementary table 2a or 2b in the paragraph summarizing the running wheel data on page 13.

• On page 14, for both the constant velocity and increased velocity rotarod tests, it is stated that “statistical values are presented on supplementary Table 3.” However, Table 3 contains statistics for the sexual behavior tests of experiment 2. Rotarod tests should instead reference supplementary tables 2a and 2b.

• On page 15, please include a reference to Supplementary table 3, which contains statistical data for Experiment 2, within the paragraph describing the results of the sexual behavior tests.

• Placing the key on the same graph may make it easier for readers to grasp the treatment variables for the brain activation figures for Exp. 2. Currently, for figures 7, 8, and supplementary figures 1 and 2, the keys are placed on separate line graphs.

• On page 17, please specify, in the section describing the signal intensity of the olfactory bulb, that it was the “SB” 16 mg/kg group displaying higher intensity on session 10.

• On page 17, please specify, in the section describing the signal intensity of the BNST, that it was the SB 16 mg/kg group that demonstrated higher intensity in session 10 compared to 1.

• On page 18, please specify, in the section describing the signal intensity of the VMH, that it was the “SB” 16 mg/kg group displaying higher intensity on session 10.

• On page 18, please include that pituitary gland results show decreases for the SB 16 mg/kg group on session 10 compared to 1.

• On page 19, please specify, in the section describing the signal intensity of the hippocampus, that it was the SB 16 mg/kg group displaying higher intensity on session 10.

Discussion

• The abbreviation SERT-KO is not defined on page 23. For those readers unfamiliar with this term, please spell out serotonin transporter knockout.

6. PLOS authors have the option to publish the peer review history of their article (what does this mean?). If published, this will include your full peer review and any attached files.

Reviewer #1: **Yes: **Genaro Alfonso Coria-Avila

Reviewer #2: No

---

## [Author Response · Author response to Decision Letter 0]

13 Jul 2022

Response to reviewers 

General 

1.- The funding information "This research was supported by grant DGAPA, PAPIIT, UNAM, IN20652. The funders had no role in study design, data collection and analysis, decision to publish, or preparation of the manuscript" is correct and was removed from the Acknowledgments section. 

2.- Figures 6 and 11 are now cited in the text. Figures 6 in page 17; Figure 11 in page 21.

3.- The captions for the supplementary figures 1, 2 and 3 are now included at the end of the manuscript.

4.- We checked the reference list 

Reviewer 1

1.- We modify the last paragraph of the abstract to clarify that “The socio sexual circuit showed a higher signal intensity on session 5 than the reward circuit and the control groups indicating that even with sexual experience the activation of the reward circuit requires the activation of the socio sexual circuit”

2.- As suggested by the reviewer we added a paragraph in the discussion section about proceptive behaviors and possible long-term effects of MnCl2 on preceptive behaviors: “These measures of the paced mating condition are a better representation of sexual motivation than hops, darts and ear wiggling because the latter ones may or no may occur when females paced the sexual interaction (Erskine 1989) an in some cases they are reduced when females paced in comparison when they don’t pace the sexual interaction (ventura Aquino & Fernandez Guasti. Physiol Behav 120, 70-76; 20139). It is also clear that the administration of MnCl2 didn’t modify any of the sexual parameters displayed by the females across the 10 weeks of testing. The MLI and LQ remained high, the percentage of exits and the return latencies are similar to what we (Arzate et al., 2011 H&B 59, 674-680; Camacho et al., 2009 Horm & Behav 56; 410-415; Corona et al., 2011, Horm & Behav 60, 264-268) and others (Erskine 1989) have consistently described.”

3.- We now indicate the doses of the mixture in mg/kg. 

4.- In the new version we mentioned at the beginning of experiment 1 that the administration of manganese was subcutaneous. 

5.- We modified figures 1,7,8, and supplementary figures 1 and 2 for clarity. 

6.- There isn’t much information about MEMRI in other species. In the discussion we described studies in rats, mice, and grasshoppers, but we added a paragraph at the endo of the conclusion to indicate the importance of considering possible toxic effect in other species.

Reviewer 2

As suggested by the reviewer we added a paragraph at the end of the abstract and the conclusion to put in a broader context the usefulness of our study.

Introduction: 

1.- As suggested by the reviewer, we now explain the importance of “increasing MR contrast in T1 weighted images”. We added the following paragraph on page 4 “Hence, when the neurons are depolarized, the active cells increase T1 signal intensity by the accumulation of manganese. Therefore, the increase of signal intensity is related to the activation of neurons.

2.- As indicated by the reviewer, we now explicitly mentioned that we investigated female paced sexual behavior

Methods:

1.- As suggested by the reviewer we now include a rationale and references for the EB dose used in the study: “These doses were used to induced high levels of sexual receptivity and we have used them repeatedly before (34, 35)”

2.- As for the number of subjects in experiment, is not necessarily a low number for behavioral studies. In our experience in tests of sexual behavior around 9/10 animals is a good number. We wanted to have 9 subjects per group but in the 8 mg/kg group, one female died of unknown reasons. Another important reason is that the use of the MRI machine is expensive and in experiment 1 we only wanted to find the ideal dose to use in experiment 2 where we could have more animals 

3.- Regarding why we didn’t use other measures of proceptivity, is the same response to reviewer 1: we added a paragraph in the discussion section about proceptive behaviors: “These measures of the paced mating condition are a better representation of sexual motivation than hops, darts and ear wiggling because the latter ones may or no may occur when females paced the sexual interaction (Erskine 1989) an in some cases they are reduced when females paced in comparison when they don’t pace the sexual interaction (ventura Aquino & Fernandez Guasti. Physiol Behav 120, 70-76; 20139).

4.- As suggested by the reviewer in the new version we explained how percent of exits and return latencies are obtained, see sexual behavior tests on page 6. “The percentage of exits is obtained by determined the number of times the female exits the male compartment after a mount, intromission, or ejaculation in comparison to the total number of each stimulus received. While the return latencies are calculated by obtaining the mean time for the female to return to the male's chamber after an exit following a mount, an intromission, or an ejaculation”

5.- We modified the abstract to be consistent with the methods section clarifying that in both experiments subjects were injected with MnCl2 in sessions 1, 5, and 10: “In experiment 1 we evaluated the effects of two doses of MnCl2, 8 and 16 mg/kg. Subjects were injected with one of the doses of MnCl2 24 hours before the test on sessions 1, 5 and 10 and immediately thereafter scanned. Female sexual behavior, running wheel and the rotarod were evaluated once a week for 10 weeks. In experiment 2 we followed a similar procedure, but females……”

6.- The X-axis in Figure 1 is now label “Sessions”

Results:

1.- We appreciate the detailed revision of the discrepancies between what was in the text and the information presented in the supplementary material. We corrected the information in the new version. 

2.- As suggested by the reviewer we modified figures 7, 8, and supplementary figures 1 and 2, to have the keys placed in the same graph and modified the legend for clarity. 

3.- We also modified pages 17,18 and 19 to make explicit that the significant changes were observed in the SB 16 mg/kg group. 

Discussion: 

1.- We defined in the discussion serotonin transporter knockout (SERT-KO) mice. 

2.- As suggested by both reviewers we added a paragraph at the end of the conclusion to put in a wither perspective the use of MEMRI: “The MEMRI methodology can be use in different species to evaluate in the same animal brain changes associated with a particular behavior, but possible toxic effects should be considered. It is also clear, that we can now evaluate many different aspects of sexual behavior and other motivated behaviors in both males and females, by MEMRI”.

---

## [Editor Report · Decision Letter 1]

18 Jul 2022

Brain circuits activated by female sexual behavior evaluated by manganese enhanced magnetic resonance imaging

PONE-D-22-12289R1

Estimado Raúl, 

We’re pleased to inform you that your manuscript has been judged scientifically suitable for publication and will be formally accepted for publication once it meets all outstanding technical requirements.

Kind regards,

Juan

Juan M Dominguez, PhD

Academic Editor

PLOS ONE

Additional Editor Comments (optional):

Thank you for carefully addressing the suggestions and concerns expressed by Reviewers in the original version of this manuscript.
---

## [Editor Report · Acceptance letter]

22 Jul 2022

PONE-D-22-12289R1 

Brain circuits activated by female sexual behavior evaluated by manganese enhanced magnetic resonance imaging 

Dear Dr. Paredes:

I'm pleased to inform you that your manuscript has been deemed suitable for publication in PLOS ONE. Congratulations! Your manuscript is now with our production department. 

Kind regards, 

on behalf of

Dr Juan M Dominguez 

Academic Editor

PLOS ONE